# Effectiveness of PCSK9 inhibitors: A Target Trial Emulation framework based on Real-World Electronic Health Records

Giulia Barbati[1]*, Caterina Gregorio[1,2,3], Arjuna Scagnetto[4], Carla Indennidate[4], Chiara Cappelletto[4], Andrea Di Lenarda[4]

1 Biostatistics Unit, Department of Medical Sciences, University of Trieste, Trieste, Italy, 2 MOX—Modelling and Scientific Computing Laboratory, Department of Mathematics, Politecnico di Milano, Milano, Italy, 3 Aging Research Center, Department of Neurobiology, Care Sciences and Society, Karolinska Institutet and Stockholm University, Stockholm, Sweden, 4 Cardiovascular Center, Territorial Specialistic Department, University Hospital and Health Services of Trieste, Trieste, Italy

* gbarbati@units.it

**Data Availability Statement:** Data are from administrative databases of the Cardiovascular Centre of Trieste. The owner of the data is Azienda

## Abstract

Low-Density Lipoprotein (LDL) cholesterol is one of the main target for cardiovascular (CV) prevention and therapy. In the last years, Proprotein Convertase Subtilisin–Kexin type 9 inhibitors (PCSK9-i) has emerged as a key therapeutic target to lower LDL and were introduced for prevention of CV events. Recently (June 2022) the Italian Medicines Agency (AIFA) modified the eligibility criteria for the use of PCSK9-i. We designed an observational study to estimate the prevalence of eligible subjects and evaluate the effectiveness of PCSK9-i applying a Target Trial Emulation (TTE) approach based on Electronic Health Records (EHR). Subjects meeting the eligibility criteria were identified from July 2017 (when PCSK9-i became available) to December 2020. Outcomes were all-cause death and the first hospitalization. Among eligible subjects, we identified those treated at date of the first prescription. Inverse Probability of Treatment Weights (IPTW) were estimated including demographic and clinical covariates, history of treatment with statins and the month/year eligibility date. Competing risk models on weighted cohorts were used to derive the Average Treatment Effect (ATE) and the Conditional Average Treatment Effect (CATE) in subgroups of interest. Out of 1976 eligible subjects, 161 (8%) received treatment with PCSK9-i. Treated individuals were slightly younger, predominantly male, had more severe CV conditions, and were more often treated with statin compared to the untreated subjects. The latter exhibited a higher prevalence of non-CV comorbidities. A significant absolute and relative risk reduction of death and a lower relative risk for the first hospitalization was observed. The risk reduction for death was confirmed in CATE analysis. PCSk9-i were prescribed to a minority of eligible subjects. Within the TTE framework, the analysis confirmed the association between PCSK9-i and lower risk of events, aligning with findings from randomized clinical trials (RCTs). In our study, PCSK9-i provided protection specifically against all-cause death, expanding upon the evidence from RCTs that had primarily focused on composite CV outcomes.

sanitaria universitaria Giuliano Isontina (ASU GI). The authors are not allowed to share data publicly as it contains sensitive, patient information. Analyzed data are linked and anonymized before being given to the analysts. The person in charge of data control for the government is: Dr. Andrea Di Lenarda, Director of Cardiovascular Center, University Hospital and Health Services of Trieste, Trieste, Italy, [ccv@asugi.sanita.fvg.it]. Data can be requested for researchers who meet the criteria at [sri@asugi.sanita.fvg.it], SC Ricerca e Innovazione Clinico Assistenziale (ASU GI), Via Giovanni Sai 1 - 3, 34128 Trieste, Italy.

**Funding:** This work was supported by MA Provider S.r.l.

**Competing interests:** The authors have declared that no competing interests exist.

# 1. Introduction

Hyperlipidemias represent one of the most important causal factors of early manifestations of atherosclerosis and organ damage, such as acute myocardial infarction (AMI), cerebral stroke and peripheral vascular disease. These conditions may depend on the presence of a primary (genetic) dyslipidemia—as in the case of familial hypercholesterolemia (FH)—or secondary to another disease, but they may also describe non-clinical situations in which the concentration of plasma lipids is inappropriately high compared to the overall cardiovascular (CV) risk of the patient. In the past, low-density lipoprotein (LDL) cholesterol was considered the main determinant of CV risk, whereas now more attention is being paid to the overall CV risk of each patient (low, moderate, high, very high), defined by a number of clinical conditions such as documented atherosclerotic cardiovascular disease (ASCD), diabetes mellitus with target organ damage and chronic kidney disease [1]. Risk scores (SCORE2, SCORE2-OP) have been developed to determine the strength required to manage all of the CV risk factors [2]. In particular cholesterol levels are one of the most important targets to manage because of their independent effect on cardiovascular outcomes and their cumulative effect on ASCD progression [3]: specific cholesterol concentration targets have been set for each risk class.

Appropriate management of dyslipidemia is one of the cornerstones of primary and secondary prevention of CV diseases, with clinical evidence of the effectiveness of the "the lower the better" therapeutic strategy, i.e. the lower the blood lipid levels, the higher the clinical benefit [4].

In the last decades, increasingly effective therapies have been proposed to reduce the levels of circulating lipids in an effective, rapid and lasting way. The most commonly approach involves the use of statins of various efficacies, which may be prescribed in monotherapy or in association with ezetimibe. Recently, a protein called Proprotein Convertase Subtilisin Kexin 9 (PCSK9) has been identified and studied for its therapeutic perspectives [5].

Inhibition of PCSK9 by the use of monoclonal antibodies has been demonstrated to significantly reduce LDL values (Low-Density Lipoprotein) by 50–70%, regardless of the therapeutic background in which it is implemented (monotherapy or in combination with the standard Lipid-Lowering Therapy, LLT) [6]. Furthermore, subjects treated with PCSK9-inhibitor (PCSK9-i) reach not only a larger reduction of LDL, but also a low rate of adverse events such as myalgia or liver or neurocognitive problems [7].

Randomized Controlled Trials (RCTs) have demonstrated the efficacy of this drug on reduction of the composite end-point of CV death, AMI, stroke, hospitalization for unstable angina, or coronary revascularization [8–10]. Since 2017, Italy has initiated the reimbursement of PCSK9-i. This decision was made by the Italian Medicines Agency (AIFA) following the 2016–2019 ESC/EAS guidelines, which set the criteria for eligibility for reimbursement [11–14].

The twofold aim of this study was twofold: 1) to utilize electronic health records (EHR) from the Observatory of CardioVascular Diseases (OCVD) in the Friuli-Venezia Giulia region (FVG, Italy) [15] to estimate the prevalence of patients eligible for PCSK9-i treatment; 2) to evaluate the Real-World Effectiveness (RWE) comparing eligible individuals treated with PCSK9-i or not.

# 2. Methods

The OCVD systematically collects integrated administrative and cardiological clinical data of residents in the Trieste and Gorizia area in Friuli-Venezia-Giulia (FVG) region in Italy (366.732 inhabitants). In Italy, all citizens have equal access to health care provided by the National Health Service (NHS). Data sources relevant for the present research are the Registry

of Births and Deaths, Hospital Discharge data, Public Drug Distribution System, the cardiological electronic e-chart, and a noteworthy feature of the FVG system, namely, the examination results from all public laboratories operating within the area. According to the current Italian law, the Regional Ethics Committee FVG (CEUR) approved the study (Protocol ID 185_2022). Data are linked and fully anonymized before being given to the analysts. In the Target Trial Emulation (TTE) approach [16], the initial step involves creating a RCT protocol. Subsequently, a protocol for analyzing observational data is developed to simulate the trial. Accordingly, we first developed the trial protocol described in Section 2.1. The emulated trial is reported in Section 2.2 and the statistical analysis in Section 2.3.

## 2.1 The targeted RCT

For the target trial, we considered the eligibility criteria for the current use of PCSK9-i in Italy (Table 1). The recruitment period spans from July 1, 2017 (when clinicians were formally allowed to prescribe PCSK9-i) to December 31, 2020. After recruitment, every eligible participant was randomly assigned to one of the two treatment arms. The first treatment arm corresponded to continue standard LLT (i.e. "Non-treated" with PCSK9-i) while the other arm was assigned to start PCSK9-i in addition to standard LLT therapy ("Treated" with PCSK9-i). Each participant was followed until death, loss to follow-up, or end of the study (December 31, 2021). The effect of interest was intention-to-treat and the primary outcome was death for all cause while the secondary outcome was the first hospitalization. Moreover, in addition to the Average Treatment Effect (ATE), the Conditional Average Treatment Effect (CATE) [17] was estimated to evaluate the absolute reduction of the risk of events in the mutually exclusive subgroups derived from the eligibility criteria as follows:

- Documented AtheroSclerotic CardioVascular event (ASCVD) as the only eligibility criteria ("ASCVD")

- Diabetes with Target Organ Damage (TOD) or at least a Risk Factor (RF) among smoking or hypertension in absence of documented ASCVD ("Diabetes TOD/RF")

- Diabetes with TOD or at least a Risk Factor (RF) in presence of documented ASCVD ("Diabetes TOD/RF + ASCVD")

- Familiar Hypercholesterolemia (FH) without Diabetes TOD/RF or documented ASCVD

The main measure of effect was the cause-specific cumulative hazard estimated at 12, 48 and 60 months. In addition, also relative risk measures (Hazard Ratios) were reported.

## 2.2 The emulated RCT

The emulated trial was designed largely in line with that described above (Table 1 and Fig 1). First, we identified a cohort of subjects with at least one available LDL determination in the recruitment period from the OCVD. For each identified LDL, the presence of any of the eligibility criteria was determined from the observational EHR as reported in Table 1 and S1 Table. In order to evaluate previous and ongoing LLT as well as statin intolerance condition [18], we used purchases from the Public Drug Distribution System (S2 Table). It is important to note that we did not require multiple measurements of LDL levels; instead, a single measurement above the threshold was considered valid. This limitation is due to the restriction of our data sources, which only include LDL tests conducted in public laboratories. LDL exams performed in private laboratories were not accessible, unless reported by the patient during a cardiological visit and included in the cardiological e-Chart. After the identification of eligible subjects, prescriptions of PCSK9-i with the date of first prescription were identified in the study time-

**Table 1. Eligibility criteria for the target trial and the emulated trial using EHR data.**

| Criteria | | Target Trial | Emulated Trial |
|---|---|---|---|
| Fast Track | Age | 18–80 years | 18–80 years |
| | Diagnoses | Recent Acute Myocardial Infarction (AMI) (in the last 12 months) or multiple cardiovascular (ASCVD) events | As target trial using Hospital Discharge data |
| | LDL | a single measurement of LDL ≥ 70 mg/dl | a single measurement of LDL ≥ 70 mg/dl |
| | Therapy | - | - |
| Secondary Prevention | Age | 18–80 years | 18–80 years |
| | Diagnoses | AtheroSclerotic CardioVascular event (ASCVD) in the history (coronary artery bypass graft, stroke/TIA, angioplasty, coronary revascularization, carotid revascularization, peripheral arterial disease, diagnosis of ischemic heart disease)<br>OR<br>Diabetes mellitus (DM) with Target Organ Damage (TOD) (i.e. microalbuminuria, retinopathy, neuropathy or renal insufficiency)<br>OR<br>DM with at least one risk factor (RF) (smoking, hypertension) | For ASCVD: as target trial using Hospital Discharge data.<br>For DM with TOD and/or RF:<br>• clinical diagnosis in the cardiological electronic e-chart<br>• drug purchases (such as DPP4i, GLP1RA, Insulin, Metformin, Repaglinide, SGLT2 inhibitors, sulfonylureas)<br>• laboratory values (glycated hemoglobin levels ≥ 6.5%), hospitalizations related to diabetes with TOD (microalbuminuria, retinopathy, neuropathy, renal insufficiency)<br>• Smoking and hypertension from diagnoses in the cardiological electronic e-chart and/or Hospital Discharge data. |
| | LDL | Three consecutive determinations performed at different times (at least 2 months apart) ≥ 70 mg/dl | A single measurement ≥ 70 mg/dl. |
| | Therapy | At least 6 months with high efficacy statin plus ezetimibe or with demonstrated intolerance | As target trial using purchases from the Public Drug Distribution System** |
| Heterozygous Familiar Hypercholesterolemia | Age | 18–80 years | 18–80 years |
| | Diagnoses | Heterozygous Familiar Hypercholesterolemia | A case of FH was identified by the presence of either an LDL ≥ 190 mg/dl (considering the "theoretical LDL" if the subject was under statin therapy, which accounts for the statin's LDL reduction power) or a total cholesterol value ≥ 310 mg/dl associated with a history of premature ASCVD at a young age, i.e. before the age of 55 for men and 60 years for women. In addition, in the case of individuals born in the FVG Region, where it is feasible to identify their first-degree relatives (data available from 1989 onwards), we identified subjects whose parents experienced an ASCVD event at a young age. Furthermore, among those who underwent at least one cardiological evaluation, the diagnosis of family history for early ASCVD events was checked.°° |
| | LDL | Three consecutive determinations performed at different times (at least 2 months apart) ≥ 130 mg/dl | A single measurement of LDL ≥ 130 mg/dl |
| | Therapy | At least 6 months with high efficacy statin plus ezetimibe or with demonstrated intolerance | As target trial using purchases from the Public Drug Distribution System** |
| Homozygous Familiar Hypercholesterolemia | Age | 18–80 years | N/A: Homozygous FH cannot be determined from available data sources |
| | Diagnoses | Homozygous Familiar Hypercholesterolemia | |
| | LDL | - | |
| | Therapy | - | |

**To derive statin intolerance we used the following criteria reported in [15]: a subject is considered as intolerant if he/she had not purchased any statins in the preceding 6 months w.r.t. the index date, but had previously made purchases of at least two types of statins (with different ATC codes or dosages), including at least one at a low dose (such as pravastatin ≤ 20 mg, simvastatin ≤ 10 mg, lovastatin ≤ 20 mg, fluvastatin ≤ 40 mg, atorvastatin ≤ 10 mg, rosuvastatin ≤ 5 mg). These purchases were tracked starting from the first availability of data on drug purchases, which was from 1995 onward.

°° The FH definition has been proposed by the Italian Society of General Practitioners (SIMG) in collaboration with the Italian Society for the Study of Artheriosclerosis (SISA) to be easily applied by GPs in their daily practice. It has been further adopted by the AIFA to regulate reimbursement of LLT in Italy.

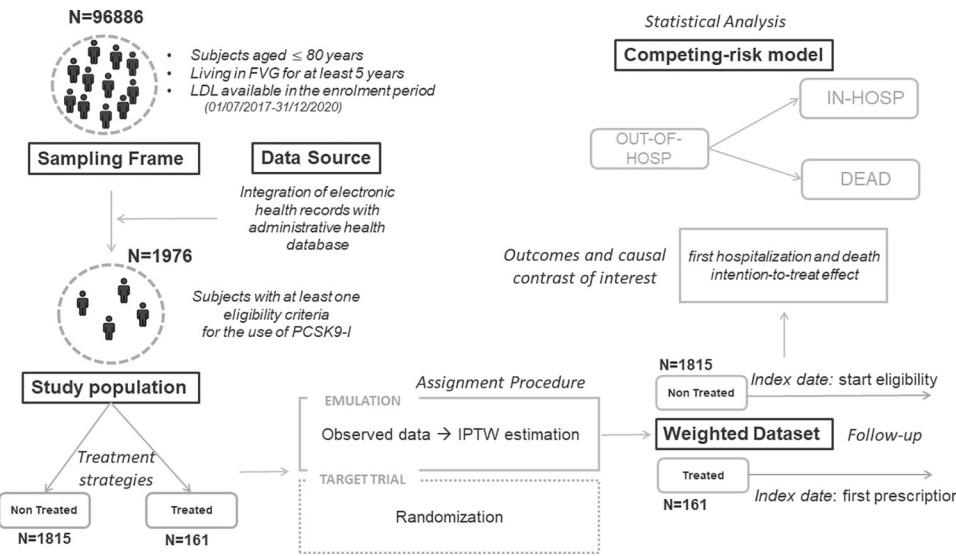

**Fig 1. TTE flow chart.**

window ("Treated"). In the cohort thus defined, each subject could have met the eligibility criteria multiple times. The first date that satisfied at least one of the eligibility criteria was identified and retained as index date for "Non treated", while for Treated subjects the index date was set to the date of the first prescription. The administrative date of censoring was set at December 31, 2021. Data extraction took place on December 30, 2022. The analyzed data underwent a process of linking and anonymization before being provided to the analysts.

## 2.3 Statistical analysis

Given the observational nature of the setting and the aim to assess treatment effectiveness, the propensity score approach was employed. This involved estimating Inverse Probability of Treatment Weights (IPTW) by considering various factors such as demographics, comorbidity load (Charlson index), documented ASCVD, diabetes with TOD or RF, history of statin treatment (including duration and adherence), the month/year of eligibility, fast-track enrollment status, and past occurrences of non-ST and ST-segment elevation Acute Myocardial Infarction (AMI-NSTEMI, AMI-STEMI). This approach is designed to emulate the assignment of study groups in a randomized trial. The inverse of the propensity score is used to assign weights to individuals, in this way adjusting for the differences in covariate distributions between the treated and untreated groups, and creating a weighted cohort where the treatment assignment is independent of the observed covariates. The list of covariates was chosen according to clinical relevance. A multivariable gradient boosting classifier algorithm as implemented in the *twang* R package was used to estimate the weights. The IPTW procedure was repeated twice, the first time including all the above cited confounders to estimate the Average Treatment Effect (ATE), and the second time excluding the variables related to the subgroups of interest, to estimate the Conditional Average Treatment Effect (CATE).

Competing risk models for the transition towards death and the first hospitalization were fitted using the Royston-Parmar flexible parametric hazard models [19] using the *flexsurv* R package [20].

The baseline transition hazard was modelled via natural cubic splines. For the selection of the number of degrees of freedom of the baseline transition hazard, the Akaike Information

Criterion (AIC) and the Bayesian Information Criterion (BIC) were used. Moreover, the overall goodness-of-fit of the models for the transition-hazards was checked by comparing the predicted values of the cumulative hazard from the models to the non-parametric estimates.

Considering the observational nature of the study, a sensitivity analysis was conducted to assess the potential influence of unmeasured confounding factors. The E-Value method, as suggested by [21], was employed for this analysis.

## 3. Results

We extracted data related to 96,886 subjects in the enrolment period, aged $\leq 80$ years, with one measure of LDL available. Among these, at least one of the eligibility criteria occurred for 1,976 subjects (2%). Among those 1976 subjects actually eligible to PCSK9-i, 1,415 (72%) presented a diabetes with TOD or RF and 1,016 (51%) patients had a documented ASCVD event (Table 2). 538 subjects (27%) presented with both conditions. 849 subjects meet the Fast Track criterion, i.e. they had an AMI in the 12 months prior to the index date or at least two ASCVD events in the history. Finally, 83 subjects presented with FH condition in primary prevention. About completeness of the data, variables corresponding to the diagnoses of interest were constructed by aggregating data from multiple sources, including cause-specific hospitalizations prior to the index date, diagnoses reported during cardiology visits, purchases of specific drugs, and specific laboratory exam values (Table 1). This aggregated information was then used to create binary variables indicating the presence or absence of a diagnosis as of the index date. As a result, there were no missing data for the diagnosis variables. For the continuous parameters in this study, there were no missing values in key variables such as age and LDL levels, since having at least one LDL measurement during the study period was an inclusion criterion for the cohort. However, there were some missing data for other laboratory values, such as Hemoglobin (6%) and Creatinine/GFR (14%). Since these variables were not used in the Propensity Score estimation, we did not apply any statistical imputation methods to replace the missing values.

Among patients satisfying clinical criteria for PCSK9-i prescription the Non treated group comprised 1815 subjects; the Treated were 161 (8% of the eligible subjects). Of note, at the time of enrolment in the study the eligibility criteria included a less broad population of patients in secondary prevention with LDL levels $\geq$100 mg/dl (in our case corresponding to 1272 subjects). Treated subjects were slightly younger, prevalently males, with more severe CV conditions and a higher rate of statin treatment with respect to the Non treated subjects. Treated subjects had higher rates of ASCVD events in their history. Non treated patients showed higher prevalence of comorbidities such as diabetes, chronic obstructive pulmonary disease (COPD) and renal diseases. History of LLT and adherence to LLT were higher in the Treated group, and it could be noted that about 1/3 of the Non Treated did not follow any statin therapy in the 6 months before the enrolment (Table 2).

### 3.1 Confounding adjustment (IPTW) and competing risk models

Results of the IPTW procedure are shown in Tables 3 and 4: for all the covariates used to estimate weights a satisfactory balance has been achieved, therefore differences in outcomes could be reasonably attributed to the effect of PCSK9-i, with the usual warning of possible unmeasured confounding. Subjects with estimated propensity scores indicating potential violations of the positivity assumption were excluded from the analysis concerning the ATE estimation. In a median follow up of 33 months (IQR 23–43), 98 deaths and 256 hospitalizations were observed. We assumed that the effect of the treatment was independent from time, since we had not sufficient statistical power to evaluate the presence of a time-varying effect. From this

**Table 2. Study population characteristics.**

| | Non treated N = 1,815 | Treated N = 161 | p-value |
|---|---|---|---|
| age, mean (SD) | 68 (9) | 65 (10) | <0.001 |
| Sex | | | 0.021 |
| F | 848 (47%) | 60 (37%) | |
| M | 967 (53%) | 101 (63%) | |
| BMI | 27.0 (25.0, 31.0) | 27.0 (25.0, 30.0) | 0.086 |
| Charlson index | 2.00 (1.00, 4.00) | 2.00 (1.00, 4.00) | 0.2 |
| Stroke | 175 (9.6%) | 10 (6.2%) | 0.2 |
| COPD | 410 (23%) | 22 (14%) | 0.009 |
| Renal Disease | 376 (21%) | 23 (14%) | 0.051 |
| Obesity | 354 (20%) | 20 (12%) | 0.028 |
| Smoke | 245 (13%) | 27 (17%) | 0.2 |
| Hypertension | 1,632 (90%) | 137 (85%) | 0.055 |
| Documented ASCVD | 892 (49%) | 124 (77%) | <0.001 |
| Diabetes | 1,596 (88%) | 117 (73%) | <0.001 |
| Diabetes TOD | 464 (26%) | 24 (15%) | 0.003 |
| **Subgroups** | | | |
| ASCVD | 442 (24%) | 36 (22%) | <0.001 |
| Diabetes TOD / RF | 865 (48%) | 12 (7.5%) | |
| Diabetes TOD / RF + ASCVD | 450 (25%) | 88 (55%) | |
| FH | 58 (3%) | 25 (16%) | |
| **CV history** | | | |
| Fast Track | 744 (41%) | 105 (65%) | <0.001 |
| Previous NSTEMI | 193 (11%) | 31 (19%) | <0.001 |
| Previous STEMI | 243 (13%) | 47 (29%) | <0.001 |
| Previous PTCA | 357 (20%) | 76 (47%) | <0.001 |
| Previous CABG | 151 (8.3%) | 33 (20%) | <0.001 |
| PAD | 238 (13%) | 32 (20%) | 0.017 |
| **Laboratory values** | | | |
| Hemoglobin (g/dl) | 13.90 (12.70, 14.90) | 14.00 (13.03, 14.90) | 0.3 |
| LDL (mg/dl) | 114 (88, 148) | 136 (107, 173) | <0.001 |
| Cholesterol (mg/dl) | 196 (166, 234) | 232 (176, 275) | <0.001 |
| Triglycerides (mg/dl) | 123 (91, 166) | 126 (93, 173) | 0.6 |
| Creatinine (mg/dl) | 0.90 (0.76, 1.07) | 0.94 (0.78, 1.06) | 0.4 |
| GFR (CKD_EPI) (ml/min/1.73 m$^2$) | 78 (63, 90) | 80 (66, 91) | 0.2 |
| **LLT history in the 6 months before enrolment** | | | |
| Any LLT therapy | 1,241 (68%) | 140 (87%) | <0.001 |
| LLT high intensity (LLT power $\geq$ 50%) | 584 (32%) | 23 (14%) | <0.001 |
| LLT low-intermediate intensity (LLT power < 50%) | 652 (36%) | 113 (70%) | <0.001 |
| No Statin therapy/intolerance | 579 (32%) | 25 (16%) | <0.001 |
| **Previous history of LLT** | | | |
| PDC | 8% (2%-22%) | 12% (4%-30%) | 0.007 |
| Time on LLT (years) | 9.7 (4.6–15) | 13 (7–19) | <0.001 |
| **Others CV therapies at enrolment** | | | |
| Antihypertensives | 1,510 (83%) | 127 (79%) | 0.2 |
| Beta-blockers | 829 (46%) | 90 (56%) | 0.013 |
| Mineralcorticoid Receptor Antagonis | 120 (6.6%) | 9 (5.6%) | 0.6 |

*(Continued)*

**Table 2.** (Continued)

| | Non treated<br>N = 1,815 | Treated<br>N = 161 | p-value |
|---|---|---|---|
| Antiplatelet agents | 1,045 (58%) | 116 (72%) | <0.001 |

For categorical variables absolute frequencies and percentages are reported. For continuous variables, median and IQR values are reported, except for Age.

ASCVD = AtheroSclerotic CardioVascular event; COPD = Chronic Obstructive Pulmonary Disease; PAD = Peripheral Arterial Disease; LLT = lipid lowering therapy; Diabetes TOD = Diabetes with Target Organ Damage; RF = Risk Factor; PDC = Percentage of Days Covered by LLT therapy (from the first recorded LLT prescription to the index date).

model, in terms of relative risk, we observed a significant relative risk reduction for the Treated group for the first hospitalization (HR = 0.78, 95% CI 0.63–0.98) and a stronger relative reduction of risk of all-cause death (HR = 0.14, 95% CI 0.07–0.27) (Table 5). According to the E-value method, the unmeasured confounding would have to be associated with a 7-fold increase in the risk of death and of the treatment to explain away the observed HR.

Concerning cause-specific hazard for the first hospitalization estimated for the ATE (Fig 2), we observed a difference of -0.02 at 12 months, -0.05 at 48 months and -0.06 at 60 months (difference in cause-specific hazards between Treated vs Non Treated), but corresponding confidence intervals around estimates are partially overlapping, indicating that when accounting for the baseline hazard in the population the absolute treatment effect is lower with respect to the relative risk reduction. For all-cause death, we observed respectively a difference of -0.02 at 12 months, -0.06 at 48 months and -0.07 at 60 months, and in this case, confidence intervals are well separated.

For the CATE analysis we estimated the effect of the treatment conditionally on the subgroups of interest (Table 6). We still obtained a significant relative risk reduction of all-cause death, independent of the subgroup considered, (HR = 0.22, 95% CI 0.15–0.33). According to the E-value method, a magnitude of 5 for the unmeasured confounding would be necessary to fully explain the estimated hazard ratio.

A trend towards a relative risk reduction for the first hospitalization was estimated (HR = 0.88, 95% CI 0.74–1.06). Interestingly, the FH group (reference for the relative risk) showed an increased relative risk of first hospitalization w.r.t. Diabetic patients with TOD/RF,

**Table 3. Propensity score diagnostics for the ATE effect estimation.**

| Variable | Standardized effect size<br>Unweighted Dataset | p-value for unbalance<br>Unweighted Dataset | Standardized effect size<br>Weighted Dataset | p-value for unbalance<br>Weighted Dataset |
|---|---|---|---|---|
| Age | -0.136 | 0.159 | 0.076 | 0.554 |
| Sex | 0.129 | 0.153 | -0.058 | 0.718 |
| Charlson index | -0.003 | 0.973 | -0.040 | 0.770 |
| Documented ASCVD | 0.324 | <0.001 | -0.219 | 0.195 |
| Diabetes with TOD or RF | 0.000 | 0.997 | 0.153 | 0.202 |
| Time on statins | 0.067 | 0.449 | -0.028 | 0.773 |
| PDC statins | 0.065 | 0.477 | -0.132 | 0.280 |
| Recent AMI or multiple CV | 0.259 | 0.003 | -0.199 | 0.171 |
| AMI NSTEMI | 0.176 | 0.092 | -0.040 | 0.681 |
| AMI STEMI | 0.381 | <0.001 | -0.030 | 0.750 |
| Year of enrolment | -0.066 | 0.471 | -0.127 | 0.467 |

**Table 4. Propensity score diagnostics for the CATE effect estimation.**

| Variable | Standardized effect size Unweighted Dataset | p-value for unbalance Unweighted Dataset | Standardized effect size Weighted Dataset | p-value for unbalance Weighted Dataset |
|---|---|---|---|---|
| Age | -0.243 | 0.008 | -0.077 | 0.408 |
| Sex | 0.129 | 0.123 | 0.098 | 0.312 |
| Charlson index | -0.069 | 0.420 | 0.000 | 0.998 |
| Time on statins | 0.214 | 0.012 | -0.035 | 0.675 |
| PDC statins | 0.139 | 0.112 | -0.082 | 0.275 |
| Year of enrolment | -0.066 | 0.413 | -0.064 | 0.525 |

and a similar relative risk of hospitalization w.r.t. patients with ASCVD. Concerning all-cause death, no significant differences in relative hazards were observed among subgroups under study.

About the cause-specific hazard for the first hospitalization in the different groups for the CATE (Fig 3), the absolute largest effects were found in the Diabetes TOD/RF +ASCVD group, ranging from -0.02 at 12 months to -0.07 and -0.08 respectively at 48 and 60 months. Then, for the FH and ASCVD group (from -0.02 at 12 months to -0.07/-0.06 at 60 months). The smaller effect was observed in Diabetes TOD+RF group (from -0.004 at 12 months to 0.02 at 60 months). These values align with clinical risk profiles assigned to each of these distinct groups. It is worth noting that confidence intervals surrounding estimates also partially overlap, as previously observed for confidence interval of the hazard ratio. For all-cause death, we observed a quite homogeneous absolute effect across groups, ranging from -0.02 at 12 months, to an interval between -0.05 and -0.07 at 48 months and finally in an interval from -0.07 to -0.09 at 60 months. All of confidence intervals were separated, indicating statistical significance.

## 4. Discussion

One of the primary aim of this study was to determine the prevalence of patients currently eligible in a specific area of Italy for PCSK9-i treatment using EHR. While these data sources have gained increasing importance in recent decades [22, 23], identifying the target population and relevant variables in order to design real-word based analyses is still a challenging task. The ability to undertake this task was made possible by the systematic integration of clinical data from the cardiological e-chart with administrative health data within the OCVD. To the best of our knowledge, this is a unique case in Italy.

Eligibility criteria for PCSK9-i prescription identify very high risk patients. We found that 2% of the initial population was eligible. To the best of our knowledge, this is the first instance where the prevalence of eligible patients for PCSK9-i has been estimated using EHR. Hence, it is not currently possible to compare this result with other findings in the existing literature.

In our opinion, this percentage likely underestimates the real population at very high risk of CV events and this result have to be interpreted in light of limitation of real-word data sources. For example, defining Familial Hypercholesterolemia (FH) was particularly complicated due

**Table 5. Competing risk model for the ATE effect.**

| Outcome | Variable | Estimate | 95% CI |
|---|---|---|---|
| First Hospitalization | Treated vs Non treated | -0.241 | -0.469; -0.014 |
| Death | Treated vs Non treated | -1.976 | -2.668; -1.285 |

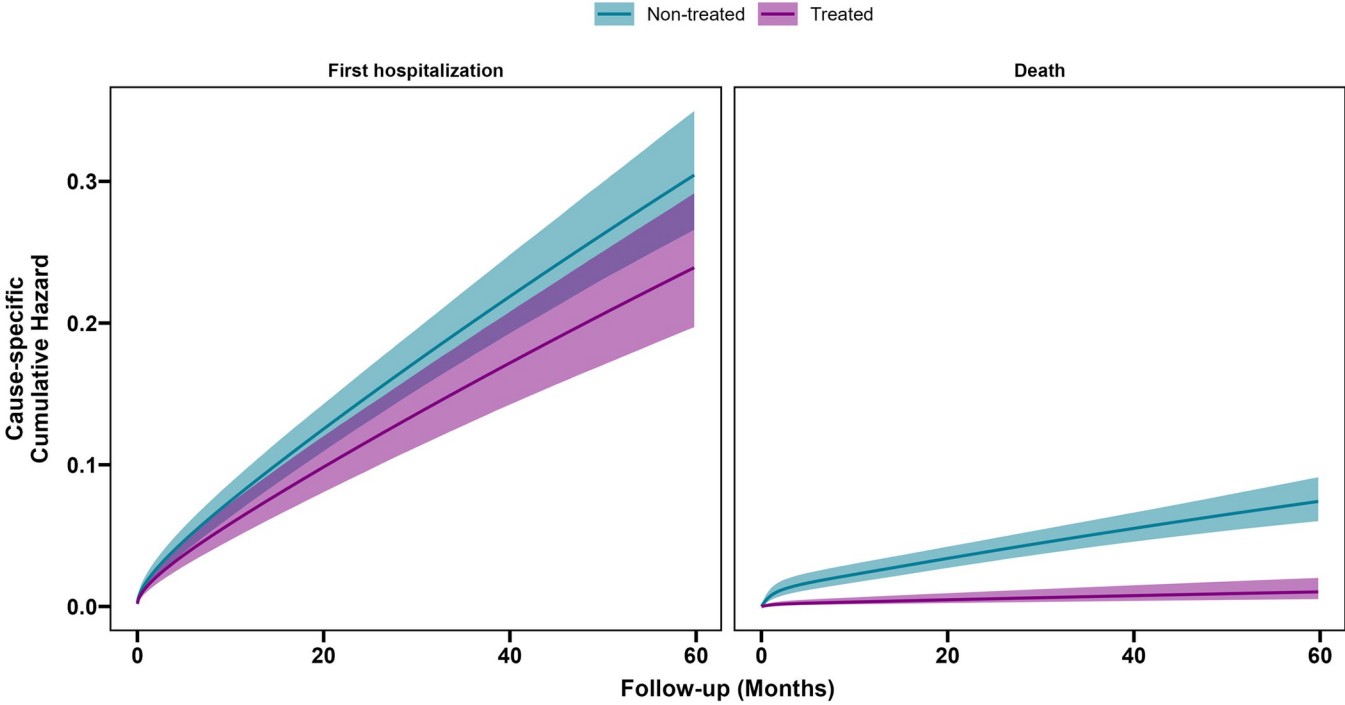

**Fig 2. Cause-specific cumulative hazard curves estimated for the ATE effect.**

to the lack of family cluster evidence for all recruited subjects. However, it is noteworthy that the prevalence of heterozygous FH obtained (4%) in relation to the study population of eligible patients aligns with recent estimates of this condition in Italy [24].

More than two third of eligible patients were affected by diabetes with RF or TOD and half of them were affected by ASCVD. Even though more than two third of them were on any LLT, the majority of subjects of the study cohort was far from the LDL target for their specific risk class (LDL first quartile in Non-treated group was 88 mg/dl and in treated was 107 mg/dl). This observation was in line with international data on the low prevalence (generally, <25%) of patients who reach the LDL target proposed by the current guidelines according with their risk class, with very high risk group more often very far from the target [25].

The proportion of individuals treated with PCSK9-i was 8% among eligible subjects, which is a lower rate compared to what has been reported in the literature. It is of interest to note that the decision of treating vs non-treating may have been driven by a lower burden of non-CV

**Table 6. Competing risk model for the CATE effect.**

| Outcome | Variable | Estimate | 95% CI |
|---|---|---|---|
| First Hospitalization | Diabetes TOD/RF vs FH | -0.577 | -1.006; -0.148 |
| First Hospitalization | ASCVD vs FH | 0.326 | -0.074; 0.726 |
| First Hospitalization | Diabetes TOD/RF+ASCVD vs FH | 0.812 | 0.436; 1.189 |
| First Hospitalization | Treated vs Non treated | -0.118 | -0.295; 0.059 |
| Death | Treated vs Non treated | -1.519 | -1.932; -1.106 |
| Death | Diabetes TOD/RF vs FH | 0.669 | -0.250; 1.587 |
| Death | ASCVD vs FH | 0.133 | -0.817; 1.083 |
| Death | Diabetes TOD/RF+ASCVD vs FH | 0.632 | -0.278; 1.541 |

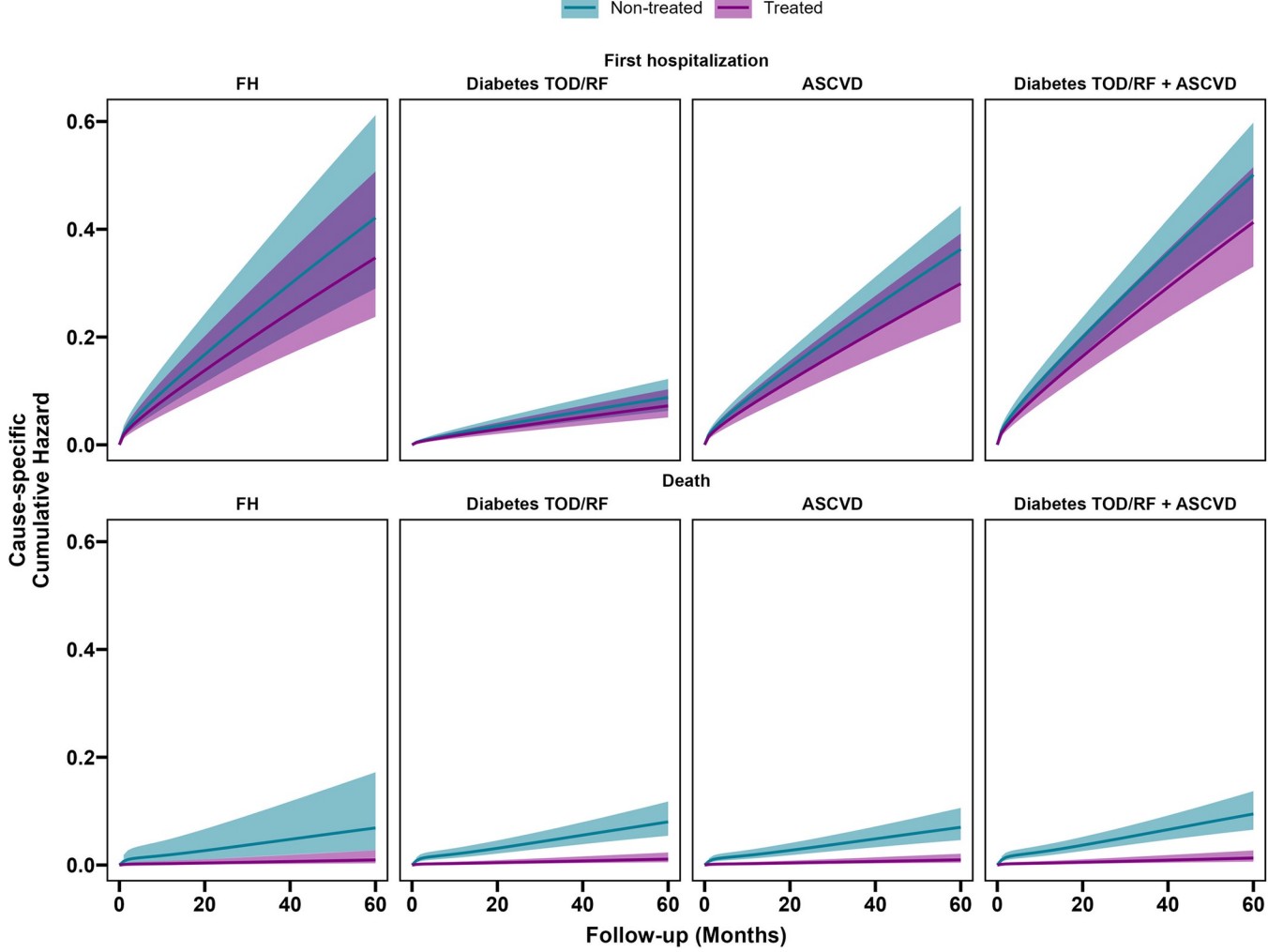

**Fig 3. Cause-specific cumulative hazard curves for death estimated for CATE effect.**

comorbidities, a slightly younger age, recurrent CV events with previous PTCA and out of target LDL concentrations despite LLT, defining an extreme high risk profile. However, it has to be underlined that the prevalence of treated patients is higher (12.7%) if considering the eligibility criteria at the time of the inclusion of subjects in the study. This percentage is in line with previous works reporting that one individual is treated for every eight eligible patients [26]. It is important to highlight that the inclusion of recently updated eligibility criteria in this study, which involved a modification of the LDL threshold, was done to align with the revised reimbursement criteria set by AIFA in June 2022 and so to provide an estimate of the prevalence updated according to the most recent rules for prescription.

In a recent study based on the regional healthcare administrative databases of the Tuscany region in Italy (3.7 million inhabitants) has been reported an incidence rate of patients on PCSK9-i in the first year of the drug's availability of 7.2 per 100,000 inhabitants [27]. The incidence rate in our study is higher, about 14 per 100,000 inhabitants; this could be probably attributed to an increase in prescriptions in the years following the first after the PCSK9-i introduction; if only the first year prescriptions are considered in our study, the incidence rate is about 11 per 100,000. Moreover, clinical and organizational barriers (number and location

of centers and physicians qualified to prescribe PCSK9-i or difficulties in the prescription procedure) could also explain regional variability of these estimates.

Concerning RWE, estimates of the relative risk reduction, in terms of the first hospitalization (ATE effect), align with the findings from randomized controlled trials. In our opinion the current study provides robust evidence regarding the reduction in the risk of death in real-word. This protective effect on mortality can be attributed to several factors. Firstly, our study population exhibits a higher cardiovascular risk profile compared to those included in clinical trials. Notably, the baseline LDL levels are, on average, more than 20 points higher than those observed in randomized controlled trials. Secondly, due to the eligibility criteria applied in Italy, our study cohort presents a higher prevalence of comorbidities than individuals enrolled in RCTs, particularly in terms of incidence of diabetes (about 28% in ODYSSEY and 36% in FOURIER versus >80% in our cohort). Moreover, our follow-up period is longer compared with the average observation period of RCTs. After the RCTs on PCSK9-i, several meta-analysis and registries-based studies [28, 29] demonstrated the cardiovascular and all-cause mortality reduction associated with PCSK9 inhibitors. From our sensitivity analysis, a hypothetical scenario in which the presence of unmeasured confounders explains away the protective effect of PCSK9-I on the risk of death appears to be highly unlikely.

To be noted that, even if reporting hazard ratio estimates in the target trial emulation is a common practice for comparing results with those from RCTs, it is well known that they have limitations as effect measures [30]. In terms of causal effects, absolute difference in cause-specific hazards is the most reliable measure. In this regard, it is important to note that assuming a constant relative treatment effect does not necessarily imply constant CATE effects. Even with a constant relative treatment effect, the treatment's impact on the absolute risk scale can vary depending on patient's baseline risk. Indeed, we observed significant variations in the association between events and treatment among subgroups of interest, particularly regarding the cause-specific hazard associated with the first hospitalization.

## 4.1 Study limitations

Some limitations of the present study need to be acknowledged to better interpret the results.

Homozygous FH in the study population was not identifiable. This represent the only criterion for PCSK9-i prescription that was not possible to derive from EHR sources. However, the frequency of this form of hypercholesterolemia has been estimated as one subject per million inhabitants [24], therefore in our opinion it should not influence the results relevantly.

Evidence from the literature suggests that the cumulative exposure to LDL over time is an important determinant of risk for CV events [31]; we did not take into account the duration of LDL exposure in our study, we focused on the cross-sectional measurement at baseline, which determined the cohort inclusion. Future research should explore the longitudinal LDL trajectory and its variability over time to understand how these factors influence outcome risk. In the present work we did not estimate the "per protocol effect" due to the difficulty in assessing treatment discontinuation, primarily because of how dispensations of PCSK9-i are recorded in Italy; however, in a recent study in which the adherence to PCSK9-i was investigated an estimated fraction of only 7% of individuals were non-adherent to the treatment [32].

Another limitation of the present work, related to the prevalence estimate of eligible subjects, pertains to the necessity for individuals to have undergone a LDL examination in public laboratories within the enrollment period. Consequently, it is important to acknowledge that not all residents in the area were included in the study, potentially introducing selection bias. Regarding the effectiveness of the treatment, the baseline selection criteria may introduce bias if the exposure of interest and other risk factors for the outcome affect the likelihood of being

in the source population, thereby altering the associations between the exposure and those risk factors. However, using IPTW we tried to mitigate potential bias caused both by baseline selection and absence of randomization to the treatment [33].

## 5. Conclusions

In conclusion, we described and implemented methods to estimate the prevalence of subjects eligible to the PCSk9-i according to the current rules, and we emulated a randomized trial using observational data based on EHR data to evaluate their effectiveness. Our results showed that effective management of CV risk is largely suboptimal compared to the real needs of a population at very high CV risk. LDL remains a relevant biomarker and one of the major drivers to optimize LLT, although a comprehensive assessment and management of CV risk must be considered to make PCSK9-i truly effective. The observational estimates of intention-to-treat effects obtained in this study can serve as valuable complements to the existing evidence from RCTs. These findings can provide additional insights for health administrators, assisting them in making informed decisions and regulations regarding the use of this treatment in high-risk cardiovascular subjects; using data from the same cohort we recently provided indications also from the point of view of the cost-effectiveness of the treatment [34]. By considering both RCT evidence and observational data, administrators can have a more comprehensive understanding of the treatment's effectiveness and its impact on the broader population. The present study has the potential to serve as a basis for future similar research using EHR in other countries or in different regions of Italy.

## Supporting information

**S1 Table. List of the ICD9-CM codes used to select the documented ASCVD events.** (DOCX)

**S2 Table. List of the ATC codes used to select Lipid Lowering Therapies (LLT) of interest for the study.** (DOCX)

## Author Contributions

**Conceptualization:** Giulia Barbati, Andrea Di Lenarda.

**Data curation:** Giulia Barbati, Caterina Gregorio, Arjuna Scagnetto.

**Formal analysis:** Giulia Barbati, Caterina Gregorio.

**Methodology:** Giulia Barbati.

**Software:** Arjuna Scagnetto.

**Supervision:** Andrea Di Lenarda.

**Writing – original draft:** Giulia Barbati, Caterina Gregorio, Carla Indennidate, Chiara Cappelletto.

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
