## [Decision Letter · Decision Letter 0]

6 May 2024

PONE-D-24-04560Effectiveness of PCSK9 inhibitors: a Target Trial Emulation framework based on Real-World Electronic Health Records.PLOS ONE

Dear Dr. Barbati,

Thank you for submitting your manuscript to PLOS ONE. After careful consideration, we feel that it has merit but does not fully meet PLOS ONE’s publication criteria as it currently stands. Therefore, we invite you to submit a revised version of the manuscript that addresses the points raised during the review process.

**Please address the requested changes made by both reviewers.**

We look forward to receiving your revised manuscript.

Kind regards,

Luiz Sérgio Fernandes de Carvalho, PhD, MSc, MD

Academic Editor

PLOS ONE

2. Please provide additional details regarding participant consent. If you are reporting a retrospective study of medical records or archived samples, please ensure that you have discussed whether all data were fully anonymized before you accessed them and/or whether the IRB or ethics committee waived the requirement for informed consent. If patients provided informed written consent to have data from their medical records used in research, please include this information.

This work was supported by MA Provider S.r.l

Reviewers' comments:

Reviewer's Responses to Questions

**Comments to the Author**

1. Is the manuscript technically sound, and do the data support the conclusions?

Reviewer #1: Partly

Reviewer #2: Yes

2. Has the statistical analysis been performed appropriately and rigorously? 

Reviewer #1: I Don't Know

Reviewer #2: Yes

3. Have the authors made all data underlying the findings in their manuscript fully available?

Reviewer #1: Yes

Reviewer #2: Yes

4. Is the manuscript presented in an intelligible fashion and written in standard English?

Reviewer #1: No

Reviewer #2: Yes

5. Review Comments to the Author

**Reviewer #1: **

The data are of interest but the wording and discussion are actually suboptimal. Indeed, "High cholesterol" is an old concept that doesn't make sense since HPS in 2004. The current approach is based on CVR at baseline and TOD more than on LDL value. As you know several registries have reported that the median value of LDL in ACS patients was below 110 mg/dL which not "HIGH". You never mention in the text the concept of time exposure which is crucial to interpret the data, as highlighted by FOURIER and OLE FOURIER, and several meta analyses by the CTT. Furthermore, the total mortality was reduced in ODYSSEY outcomes but the hierarchical statistical analysis didn't allow definitive conclusion. The conclusion is not separated from the discussion. Both may be largely improved with a better overview of published RCT and registries.Best regards.

**Reviewer #2: **

I read the manuscript with interest, which significantly contributes to the body of knowledge on the treatment of cardiovascular diseases with PCSK9 inhibitors, using complex analyses and real-world data.

The Target Trial Emulation approach represents a sophisticated and innovative methodological strategy that aims to simulate the rigor of randomized clinical trials within the limitations of observational studies. The use of IPTW and competitive risk models is a commendable attempt to adjust for confounders and deal with competing events. Therefore, I make my considerations:

1. Although the methodology is robust, a more in-depth discussion on the treatment of potential confounders and missing data could enrich the analysis. Understanding how these issues were addressed would help to assess the robustness of the results.

2. Was consideration given to generalizing the findings to other populations or contexts? It might be an important consideration to discuss more thoroughly the extent to which the results can be applied beyond the study sample.

6. PLOS authors have the option to publish the peer review history of their article (what does this mean?). If published, this will include your full peer review and any attached files.

Reviewer #1: No

Reviewer #2: No

---

## [Author Response · Author response to Decision Letter 0]

20 Jun 2024

Reviewer #1: 

The data are of interest but the wording and discussion are actually suboptimal. Indeed, "High cholesterol" is an old concept that doesn’t make sense since HPS in 2004. The current approach is based on CVR at baseline and TOD more than on LDL value. As you know several registries have reported that the median value of LDL in ACS patients was below 110 mg/dL which not "HIGH". You never mention in the text the concept of time exposure which is crucial to interpret the data, as highlighted by FOURIER and OLE FOURIER, and several meta analyses by the CTT. Furthermore, the total mortality was reduced in ODYSSEY outcomes but the hierarchical statistical analysis didn't allow definitive conclusion. The conclusion is not separated from the discussion. Both may be largely improved with a better overview of published RCT and registries. Best regards.

Response to Reviewer #1: 

We sincerely thank the Reviewer for His/Her insightful comments and suggestions, which have significantly contributed to improve our manuscript. We have made every effort to address all the points highlighted. In the revised version of the manuscript we have extensively modified the Abstract, the Methods and the Discussion sections in order to follow the Reviewer’s suggestions. Moreover, we have included several new references as indicated by the Reviewer and we have also added a distinct “Limitations” and “Conclusions” sections. 

We completely agree with the Reviewer that the concept of “high cholesterol” is outdated and we apologize for not addressing this clearly in the previous version. In the revised version of the manuscript, we have focused more on cardiovascular risk and on target organ damage concepts as proposed by the current guidelines and literature on this topic.

Moreover, as the Reviewer correctly pointed out, the concept of time exposure is crucial, both from the point of view of the treatment and of the LDL concentration. We used time exposure to the LLT treatment before the index date (duration and adherence), including these variables in the Propensity Score estimation in order to balance the differences between Treated and Non Treated. In the emulated trial we conducted an ITT (Intention To Treat) analyses, i.e. we assumed that subjects were following the prescribed treatment at baseline. This could be a limitation of the present study. However, it’s noteworthy that the specific drug under investigation PCSK9-i seem to exhibit remarkably low non-adherence rates in the Italian context (estimated around 7%, reference : Arca M, Celant S, Olimpieri PP, Colatrella A, Tomassini L, D’Erasmo L, et al. Real-World Effectiveness of PCSK9 Inhibitors in Reducing LDL-C in Patients With Familial Hypercholesterolemia in Italy: A Retrospective Cohort Study Based on the AIFA Monitoring Registries. J Am Heart Assoc. 2023). This is primarily attributed to the PCSK9-i’s distribution protocol in Italy, where the drug is not typically acquired through local drug stores but instead directly collected from hospitals. This unique system significantly minimizes the likelihood of non-adherence. Regarding the duration of LDL exposure in our study, as correctly observed by the Reviewer, we concentrated solely on the cross-sectional measurement at baseline, which determined cohort inclusion. Future research should explore the longitudinal LDL trajectory and its variability over time to understand how these factors influence outcome risk. 

Finally, regarding total mortality, it is true that it was not the primary end-point of RCTs but in a recent extensive meta-analysis [Reference: Imran TF, Khan AA, Has P, Jacobson A, Bogin S, Khalid M, et al. Proprotein convertase subtilisn/kexin type 9 inhibitors and small interfering RNA therapy for cardiovascular risk reduction: A systematic review and meta-analysis. PLOS ONE 2023] demonstrated the cardiovascular and all-cause mortality reduction associated with PCSK9 inhibitors. 

Despite the limitations of our study, which are now summarized in a specific section, we hope our results will enhance readers’ awareness of the high percentage of patients who are potentially eligible but remain untreated with PCSK9 inhibitors. This awareness is one of the key goals of our work. Thank you again for your valuable feedback.

Reviewer #2: 

I read the manuscript with interest, which significantly contributes to the body of knowledge on the treatment of cardiovascular diseases with PCSK9 inhibitors, using complex analyses and real-world data. The Target Trial Emulation approach represents a sophisticated and innovative methodological strategy that aims to simulate the rigor of randomized clinical trials within the limitations of observational studies. The use of IPTW and competitive risk models is a commendable attempt to adjust for confounders and deal with competing events. Therefore, I make my considerations:

1. Although the methodology is robust, a more in-depth discussion on the treatment of potential confounders and missing data could enrich the analysis. Understanding how these issues were addressed would help to assess the robustness of the results.

2. Was consideration given to generalizing the findings to other populations or contexts? It might be an important consideration to discuss more thoroughly the extent to which the results can be applied beyond the study sample.

Response to Reviewer #2: 

We sincerely thank the Reviewer for His/Her insightful comments and suggestions, which have significantly contributed to improve our manuscript. We have made every effort to address all the points highlighted. For the first issue, i.e. a more in-depth discussion on the treatment of potential confounders and missing data, we added more details about these aspects in the revised version of the statistical analysis section as follows: 

“The inverse of the propensity score is used to assign weights to individuals, in this way adjusting for the differences in covariate distributions between the treated and untreated groups, and creating a weighted cohort where the treatment assignment is independent of the observed covariates”.

However, given the observational nature of our study, it is also pertinent to acknowledge the likelihood of unmeasured confounding factors influencing our estimates. Consequently, as already reported in the previous version of the manuscript, we have taken measures to address this concern by incorporating a sensitivity analysis utilizing the EValue method, as proposed by VanderWeele and Ding (see the reference in text).

Regarding the issue of missing data, we apologize for the lack of clarity in the previous version of the manuscript. We now clarified better this point in the “Results” section as follows: 

“About completeness of the data, variables corresponding to the diagnoses of interest were constructed by aggregating data from multiple sources, including cause-specific hospitalizations prior to the index date, diagnoses reported during cardiology visits, purchases of specific drugs, and specific laboratory exam values (see Table 1). This aggregated information was then used to create binary variables indicating the presence or absence of a diagnosis as of the index date. As a result, there were no missing data for the diagnosis variables. For the continuous parameters in this study, there were no missing values in key variables such as age and LDL levels, since having at least one LDL measurement during the study period was an inclusion criterion for the cohort. However, there were some missing data for other laboratory values, such as Hemoglobin (6%) and Creatinine/GFR (14%). Since these variables were not used in the Propensity Score estimation, we did not apply any statistical imputation methods to replace the missing values”. 

Regarding the reviewer's second comment about the external validation of our findings, we believe that, given the specific characteristics of the Italian Health System, it would be possible to replicate our study in other Italian regions. This would require establishing a data linkage between regional administrative health data and clinical registries. However, generalizing our findings to other countries with different health systems and different guidelines for prescribing PCSK9 inhibitors would be challenging. 

We hope that these clarifications suggested by the Reviewer have improved the clarity and readability of our manuscript. Thank you again for your valuable feedback.

---

## [Decision Letter · Decision Letter 1]

30 Jul 2024

Effectiveness of PCSK9 inhibitors: a Target Trial Emulation framework based on Real-World Electronic Health Records.

PONE-D-24-04560R1

Dear Dr. Barbati,

We’re pleased to inform you that your manuscript has been judged scientifically suitable for publication and will be formally accepted for publication once it meets all outstanding technical requirements.

Kind regards,

Luiz Sérgio Fernandes de Carvalho, PhD, MSc, MD

Academic Editor

PLOS ONE

Additional Editor Comments (optional):

Reviewers' comments:

Reviewer's Responses to Questions

**Comments to the Author**

1. If the authors have adequately addressed your comments raised in a previous round of review and you feel that this manuscript is now acceptable for publication, you may indicate that here to bypass the “Comments to the Author” section, enter your conflict of interest statement in the “Confidential to Editor” section, and submit your "Accept" recommendation.

Reviewer #2: All comments have been addressed

Reviewer #3: All comments have been addressed

2. Is the manuscript technically sound, and do the data support the conclusions?

Reviewer #2: Yes

Reviewer #3: Yes

3. Has the statistical analysis been performed appropriately and rigorously? 

Reviewer #2: Yes

Reviewer #3: Yes

4. Have the authors made all data underlying the findings in their manuscript fully available?

Reviewer #2: No

Reviewer #3: Yes

5. Is the manuscript presented in an intelligible fashion and written in standard English?

Reviewer #2: Yes

Reviewer #3: Yes

6. Review Comments to the Author

Reviewer #2: (No Response)

Reviewer #3: (No Response)

7. PLOS authors have the option to publish the peer review history of their article (what does this mean?). If published, this will include your full peer review and any attached files.

Reviewer #2: No

Reviewer #3: **Yes: **ALEXANDRE ANDERSON DE SOUSA MUNHOZ SOARES

---

## [Editor Report · Acceptance letter]

14 Aug 2024

PONE-D-24-04560R1 

PLOS ONE

Dear Dr. Barbati, 

I'm pleased to inform you that your manuscript has been deemed suitable for publication in PLOS ONE. Congratulations! Your manuscript is now being handed over to our production team.

Kind regards, 

on behalf of

Dr. Luiz Sérgio Fernandes de Carvalho 

Academic Editor

PLOS ONE